# Reversed in Time: A Novel Temporal-Emphasized Benchmark for Cross-Modal Video-Text Retrieval

## ABSTRACT

Video-text retrieval is an important task in the multimodal understanding field. Temporal understanding makes video-text retrieval more challenging than image-text retrieval. However, we find that the widely used video-text benchmarks have shortcomings in comprehensively assessing abilities of models, especially in temporal understanding, causing large-scale image-text pre-trained models can already achieve comparable zero-shot performance with video-text pre-trained models. In this paper, we introduce RTime, a novel temporal-emphasized video-text retrieval dataset, constructed through a top-down three-step scheme. We first obtain videos of actions or events with significant temporality, and then reverse these videos to create harder negative samples. We then recruit annotators to judge the significance and reversibility of candidate videos, and write captions for qualified videos. We further adopt GPT-4 to extend more captions based on human-written captions. Our RTime dataset currently consists of 21k videos with 10 captions per video, totalling about 122 hours. Based on RTime, we propose three retrieval benchmark tasks: RTime-Origin, RTime-Hard, and RTime-Binary. We further enhance the use of harder-negatives in model training, and benchmark a variety of video-text models on RTime. Extensive experiment analysis proves that RTime indeed poses new and higher challenges to video-text retrieval. We will release our RTime benchmarks to further advance video-text retrieval and multimodal understanding research.

## CCS CONCEPTS

• **Information systems** → **Video search**; **Multimedia and multimodal retrieval**; **Evaluation of retrieval results**.

## KEYWORDS

Video Retrieval; Cross-modal Retrieval; Video-text Benchmark

## 1 INTRODUCTION

Video-text retrieval has been widely used in various real-world scenarios, such as video search engines and video recommendation systems. It is more challenging than image-text retrieval as it requires understanding the visual semantics of multiple frames not only spatially but also temporally. In recent years, the introduction of large-scale vision-language pre-trained models [8, 10, 17, 20, 28, 31, 43, 48, 49], which learn cross-modality alignment

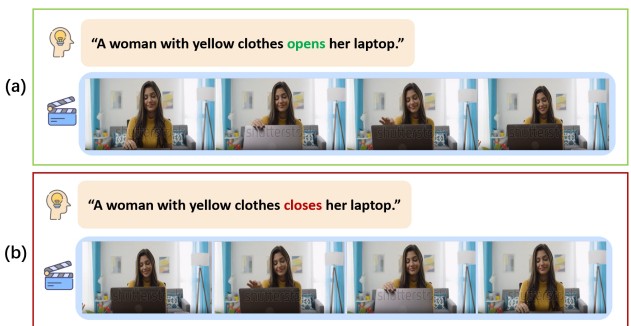

**Figure 1: Videos in (a) and (b) have identical spatial appearance but opposite temporal semantics (Open vs Close ), which can only be differentiated through temporal understanding. They are considered as temporally harder-negatives of each other.**

through contrastive learning, has brought significant performance improvements to video-text retrieval. These models can be roughly divided into two types: one type focuses on transferring image-text pre-trained models to the video domain (e.g. CLIP4Clip [38], X-Pool [14], X-Clip [39]), and the other type focuses on utilizing existing video-text datasets (e.g. HowTo100M [40], WebVid [3]) and employing diverse pre-training objectives to perform video-text pre-training, such as Frozen [3], Internvideo [53], UMT [31], Vindlu [8], Violet [11], ALPro [27], etc.

While being excited about the performance improvements achieved by recent models, we also wonder whether these models have actually significantly improved video semantic understanding capabilities, especially in terms of temporal understanding. For example, in Figure 1, the only way to differentiate the two videos with opposite temporal semantics (open laptop vs. close laptop) is through temporal understanding. Such videos with very similar spatial appearance but very different temporal semantics can be considered as temporally harder-negatives of each other. Benchmark datasets containing harder-negative samples are desired to rigorously verify the video understanding capabilities of models. However, previous works [2, 6, 25, 51, 60] point out that there is a notable lack of a video-text benchmark that emphasizes the temporal understanding. We randomly sample 100 videos from the MSRVTT [58] test set and find that only 10% of the video-text pairs involve temporal semantics[1]. Besides, most video-text datasets are created without explicitly incorporating temporally harder-negative samples, which makes them insufficient for evaluating the temporal understanding capabilities of models.

Furthermore, on the widely used video-text retrieval datasets, such as ActivityNet-Caption [24], MSR-VTT [58], and DiDeMo [1], an image-text pre-trained model [43] or a model pre-trained on

---

[1] Please refer to the supplementary material for the evaluation criteria

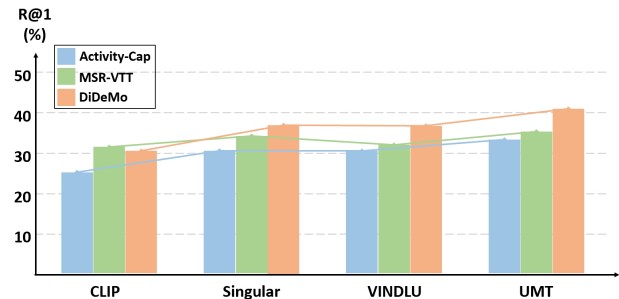

**Figure 2: Zero-shot performance of different models on some existing video-text datasets. Models without temporal understanding, such as image-text pre-trained models (e.g. CLIP) and models trained using single frames (e.g. Singular), have achieved comparable performance to models trained using video-text pairs (e.g. VINDLU and UMT), indicating that these datasets are insufficient to comprehensively validate video understanding capabilities of models, especially in terms of temporal understanding. Compared models are in the same scale.**

video-text data using a single frame [25] through simple multi-frame aggregation (e.g. mean pooling or concatenating) without temporal modeling can already achieve comparable performance to multi-frame video-text pre-trained models [8, 31], as illustrated in Figure 2. Previous work [28] also shows that BLIP could even achieve 43.3% R@1 zero-shot performance on MSR-VTT, surpassing both UMT and VINDLU. This suggests that these datasets are deficient in assessing models' retrieval capabilities, particularly in terms of temporal understanding.

To address the aforementioned deficiencies in current datasets, we propose to construct a new temporal-emphasized dataset named **RTime** and establish new benchmarks for video-text retrieval. The most prominent feature of our new dataset is its emphasis on temporal understanding, especially the inclusion of temporally harder-negative samples as exemplified in Figure 1. Specifically, we adopt a top-down three-step scheme to construct our dataset, as illustrate in Figure 3. We first brain-storm common-sense information about typical activities with strong temporality (e.g. in the format: open/close something) to form the initial activity list, then further expand it using GPT-4, followed by manual verification to ensure that each activity has its temporally reversed counterpart (harder-negative). Subsequently, we employ GPT-4 to replace the "something" in each action with typical objects, resulting in a plethora of phrases containing activities and specified objects. These phrases are then utilized as queries to search for videos on the internet through search engines, leading to the collection of a substantial amount of videos. Next, we recruit a group of professional annotators to filter and annotate the collected videos. We provide both the original and reversed video pairs to annotators and ask them to detect whether the video can be reversed in time. The annotators select videos that meet requirements and then annotate each video with fine-grained descriptions. We further apply GPT-4 to rewrite nine semantically similar sentences for each video based on the human-written caption to allow for more diverse vision-language alignment, which has been demonstrated beneficial to

vision-language contrastive learning [9]. The current version of our RTime, a fine-grained and temporal-emphasized dataset, contains 21k videos and 210k video-text pairs, totaling approximately 122 hours. Among these videos, 16,530 have their temporally harder-negative counterparts, accounting for 76.8% of the entire dataset, posing higher challenges to the video-text retrieval task.

To comprehensively assess retrieval models base on RTime, we establish three evaluation tasks: *RTime-Origin Retrieval, RTime-Hard Retrieval* and *RTime-Binary Retrieval*. RTime-Origin Retrieval is the typical video-text retrieval task, where the retrieval pool only contains the originally retrieved video-text samples. For RTime-Hard Retrieval, the reversed counterparts of videos and accompanying texts are added in the test set, which demands the model to have stronger capability to handle temporal understanding. For RTime-Binary Retrieval, given the query, the model needs to select the correct corresponding sample from the two candidate samples, where the only difference between them lies in the temporal aspect. Moreover, we evaluate the performance of several state-of-the-art models on the three video-text retrieval tasks based on RTime, and conduct empirical studies on some factors that may affect the temporal understanding capability of video-text models. Extensive experiment results show that although models pre-trained on a single frame without considering temporal information can achieve superior performance on traditional datasets such as MSR-VTT, on our RTime dataset, they significantly lag behind those models pre-trained with temporal information on multiple frames, demonstrating that our RTime indeed improves upon deficiencies in previous benchmarks and enables new temporal-emphasized video-text retrieval evaluations.

The main contributions of this work are summarized as follows:

- Through collecting videos with strong temporality and reversing them in time as harder negatives, we build RTime, a novel temporal-emphasized dataset, via a top-down three-step construction with the assistance of Large Language Models.
- Based on RTime dataset, we establish three benchmark tasks: RTime-Origin Retrieval, RTime-Hard Retrieval, and RTime-Binary Retrieval, which can more comprehensively assess the video understanding capabilities of models, especially in temporal understanding.
- We carry out extensive experiments with a variety of current state-of-the-art models and conduct empirical studies about impact factors in temporal understanding. Experimental results show that our new RTime dataset does correct shortcomings in traditional video-text datasets, and poses new and higher challenges to video-text retrieval.

## 2 RELATED WORKS

### 2.1 Video-Text Benchmark Datasets

Various video-text retrieval benchmark datasets have been proposed through collecting videos from the internet and manually annotating with captions, event timestamps, and other relevant information. For example, MSR-VTT [58] includes 10,000 video clips, with 20 manually annotated textual descriptions for each clip, making it one of the widely adopted benchmarks in the video-text retrieval and video-language understanding domain. VATEX [52] selects videos from a subset of Kinetics-600 dataset [23] and annotates

them with multi-lingual descriptions. ActivityNet-Caption [24] contains 20,000 YouTube videos, each annotated with descriptions and timestamps for events. DiDeMo [1], collected from Flickr, contains 26,892 video clips. In ActivityNet-Caption and DiDeMo, the video-text retrieval evolves into paragraph-video retrieval, where all descriptions of a video are concatenated into a single paragraph.

Additionally, some studies have recognized the limitations of widely used benchmarks in temporal evaluations and have attempted to construct benchmarks with a focus on temporal aspects. Hendricks et al. [16] concatenate clips of different events from the same video in DiDeMo [1] along with event descriptions. Lei et al. [25] reuse the Something-Something dataset [15] and propose SSV2-Label and SSV2-Template. Li et al. [32] sample videos from test set of MSRVTT [58] and VATEX [52], employed the GPT-assistant annotation framework to generate temporal counterfactual captions for the videos. In this work, we address such insufficiency in existing benchmarks and introduce a new dataset that emphasizes the temporal aspect of videos by including their harder-negative samples, the temporally reversed counterparts, using both manually and GPT-assisted data construction approach.

## 2.2 Video-Text Retrieval Methods

Cross-modal retrieval has been widely explored in previous works [5, 7, 19, 21, 33, 34, 42, 47, 54, 61]. Current video-text retrieval methods can be roughly divided into three types:

**Offline feature extraction and fusion.** Offline feature extractors are the main components commonly used in early video-text retrieval methods. For example, MMT [12] employs multiple distinct models for feature extraction and utilizes a cross-modal transformer for fusion. VideoCLIP [57] utilizes S3D [56] to extract video features and applies contrastive learning to align video and text embeddings.

**Transferring image-text pretrained models.** This type of methods utilizes pre-trained image-language models (e.g. ALBEF [29], CLIP [43], BLIP [28]) and transfers them to video retrieval tasks [14, 22, 35, 36, 38, 39]. For example, CLIP4Clip [38] leverages CLIP image encoder to encode videos frame by frame and designs modules for inter-frame information aggregation. TS2Net[35] introduces token shift and token selection modules, further enhancing the interaction of inter-frame information.

**Video-text pre-trained models**. This type of methods learns a video-text pre-trained model from large-scale video-text datasets. Various design of video encoders have been extensively explored [4, 13, 15, 18, 37, 46, 50, 59]. ClipBERT [26] pioneers the end-to-end video-text pre-training by sparsely sampling from videos. Frozen [3] adopts Timesformer [4]as video encoder for conducting joint pre-training on large-scale video-text and image-text datasets. VIN-DLU [8] investigates crucial factors in the design of video-text pre-trained models and demonstrates the importance of pre-training datasets covering video-text data. UMT [31] utilizes the CLIP image encoder as a teacher to train the video encoder, achieving state-of-the-art zero-shot performance on multiple downstream video-text retrieval datasets. Through experimental analysis on our new RTime dataset, we show that despite the success of these previous video retrieval methods on previous benchmarks, their true video understanding capabilities, especially in terms of temporal understanding, still have a lot of room for improvement.

## 3 RTIME: NOVEL VIDEO-TEXT BENCHMARK

As currently available widely-used benchmarks are insufficient to comprehensively assess the capabilities of models on video understanding, especially temporal understanding, we propose to construct a new video-text retrieval benchmark dataset to meet the higher fine-grained and temporal-emphasized evaluation requirements. Manually building a new benchmark from scratch is very expensive and time-consuming, so we leverage the power of LLMs to improve efficiency and reduce the cost of dataset construction. We put human in the verification loop[2] to control the data quality during the construction process. Specifically, we propose a top-down three-step data construction pipeline as illustrated in Figure 3, including seed activity list proposal, activity list enrichment, and video acquisition and annotation. Following this pipeline, we build a new fine-grained temporal-emphasized dataset for video-text retrieval, namely the "**reversed in time**" (**RTime**) dataset.

### 3.1 RTime Dataset Construction

To ensure the temporal emphasis and high quality of our dataset, we propose a top-down three-step data construction pipeline, which first progressively forms a comprehensive list of activities by leveraging human knowledge and world knowledge of LLMs (e.g. GPT-4). Each activity in the list may have its temporally opposite activity, so temporally harder-negatives can be constructed for each activity. We further leverage human capabilities and machine capabilities to acquire and annotate videos crawled from the internet based on the activity list. The specific steps in the pipeline are as follows.

*3.1.1* **Step 1: Seed Activity List Proposal.** By filtering labels from existing action recognition datasets [15, 23, 41] and our brainstormed activity proposals, we initiate an atomic-level activity pair list, $A_h = \{(a_i, \widetilde{a_i})\}$, each containing an activity with a pronounced temporal emphasis, as well as its temporally opposite counterpart (e.g. (open, close)). To improve the diversity of the initial list, we leverage the world knowledge of GPT-4 to suggest more activities and their temporally opposite counterparts through few-shot in-context learning. Specifically, we provide GPT-4 with a few action pairs in $A_h$ and instruct it to generate more samples. We then manually curate the list of activities, eliminating those activity pairs that are either illogical or may be indistinguishable via video. We end up with 144 activity pairs $A = \{(a_i, \widetilde{a_i})\}_{i=1}^{144}$, containing 288 verb phrases.

*3.1.2* **Step 2: Activity List Enrichment**. Directly using the activity list from step 1, which does not contain concrete objects, as queries to search for videos is not optimal. Therefore, leveraging the world knowledge and in-context learning capability of GPT-4, we prompt it to substitute [something] in each activity list with concrete objects to form a verb-noun activity list. Specifically, for each $(a_i, \widetilde{a_i}) \in A$, we instruct GPT-4[3] to generate the verb-noun phrases $L = \{(a_i + n_j, \widetilde{a_i} + n_j) \mid 1 \leq i \leq 144, n_j \in O\}$, where $O$ denotes the object set. On average, we append 20 objects to each activity, resulting in an enriched list of 5,760 diverse activities.

---

[2]Please refer to supplementary materials for details of human verification
[3]Please refer to supplementary materials for GPT-4 prompts

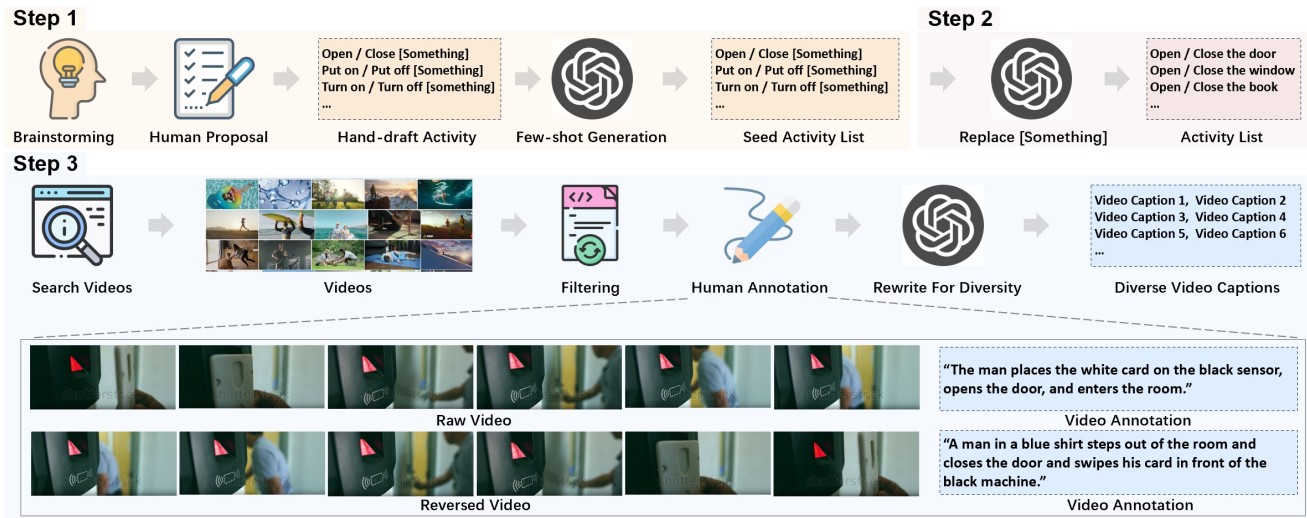

**Figure 3: Illustration of our top-down three-step dataset construction process. We first generate an action list where each action can have its meaningful temporally reversed counterpart. We then use GPT to supplement actions with objects. Videos based on the enriched activity list are crawled from the internet, followed by a filtering process to balance the label distribution. Finally, we recruit human annotators to verify the temporal information in videos and write up fine-grained descriptions. GPT is employed to rewrite based on human-written descriptions to increase caption diversity.**

3.1.3 **Step 3: Video Acquisition and Annotation.** Applying the enriched activity list as queries, we search for videos on the internet using search engines. We then go through a series of processes to filter low-quality videos, produce harder-negative samples by reversing videos, annotate videos, and rewrite annotations for diversity. We once again take full advantage of LLMs and human expertise to improve and ensure the quality of our dataset throughout the whole process.

**Raw Video Acquisition.** Directly downloading videos based on $\mathbf{L}$ is sub-optimal because many of the retrieved videos do not match the query very well due to the limited performance of the search engine. To improve the overall quality, we paid to recruit seven workers to search videos with both $(a_i + n_j)$ and $(\widetilde{a_i} + n_j)$ as queries using a video search engine. Then they filter out any activity that falls under the following conditions: 1) the activity can be identified without relying on temporal information. For example, "hold basketball" can be identified with a single static image, whereas "taking off shoes" requires temporal information. 2) the number of videos retrieved using this activity as a query is less than 50. 3) less than 50% of all the videos retrieved based on this activity correctly match this activity. After such a manual filtering process, we obtain a refined activity list $\mathbf{F} \in \mathbf{L}$, containing approximately 800 activities with strong temporal nature. To further balance the distribution of objects, we calculate the frequency of nouns in $\mathbf{F}$. For activities with lower noun frequencies, we collect top 30 videos, whereas for activities with higher noun frequencies, we collect top 20 videos. We end up collecting approximately 21,000 videos $\mathbf{V}_{raw} = \{v_i\}_{i=1}^{21,000}$ that match our requirements.

**Video Reversion.** If one wants to specifically focus on evaluating the temporal understanding ability of the model, we believe it is necessary to include harder-negative samples, that is, videos with similar visual appearance but exactly opposite temporal semantics.

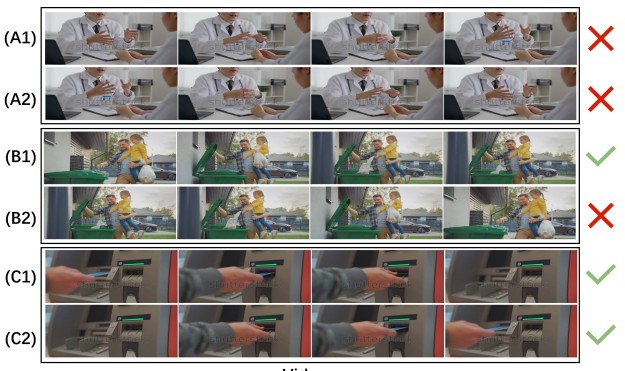

**Figure 4: Illustration of some video samples in our initial video pool. A1 and A2: temporally insignificant as there is no significant difference between the raw video and its reversed counterpart. B1 and B2: 'garbage comes out of trash can' is unreasonable in the reversed counterpart, thus only the raw video B1 is kept. C1 and C2: both are temporally significant and temporally meaningful.**

Since each activity in our list has its temporally opposite activity (e.g. open the door vs. close the door), we can reverse the raw video $v_i$ to get its harder-negative counterpart $\widetilde{v_i}$. By doing so, each video and its reversed version $(v_i, \widetilde{v_i})$ share same visual appearances but with reverse temporal order, resulting in completely opposite visual semantics. So we expand $\mathbf{V}_{raw} = \{v_i\}_{i=1}^{21,000}$ into our initial video pool $\mathbf{V} = \{(v_i, \widetilde{v_i})\}_{i=1}^{21,000}$.

**Manual Annotation.** We recruit 23 professional annotators (15 females and 8 males), who are all English majors with an average English proficiency level equivalent to a score of 7 on the IELTS,

A man in an orange hard hat watches two blue boxes pass from right to left on a conveyor belt in the workshop.

In the workshop, a man in an orange hard hat stands in front of a conveyor belt, watching two blue boxes move from left to right.

The camera focuses on a blue map of the world and gradually zooms in to focus on the map of Australia.

The camera focuses on the map of Australia and recedes until a blue map of the world is revealed.

A young Asian doctor in a blue uniform with a mask on his face is putting on a surgical hat.

A young Asian doctor in a blue uniform with a mask is taking off a surgical hat.

A person holds the light bulb with the left hand, turns the light bulb counterclockwise, and takes it off.

A man mounts an incandescent bulb on a lampshade suspended from the ceiling and turns it clockwise.

**Figure 5: Examples of some videos and their associated human-written captions from our RTime dataset. Green and red terms in the text description indicate their temporal semantic difference.**

to conduct manual annotation[4] on the initial video pool **V**. Both the original video $v_i$ and its reversed version $\widetilde{v_i}$ are provided to the annotator, who needs to first determine that the raw video $v_i$ indeed has meaningful strong temporal nature by applying the following rules: 1) it contains a distinct temporal-related activity; or 2) it contains consecutive activities with significant differences; or 3) it involves an apparent change in the state of an object; or 4) it contains observable changes in the position of an object, etc. Subsequently, the annotator needs to evaluate whether the reversed video $\widetilde{v_i}$ matches a meaningful real-world scenario, excluding unrealistic scenarios such as anti-gravity phenomena or a large number of cars driving backwards on the street. We illustrate some examples in Figure 4, where A1 and A2 have very similar semantics, so they are considered as temporally insignificant and are eliminated. B1 and B2 have different semantics, but 'garbage comes out of trash can' in B2 is unreasonable. Only B1 is kept and annotated which is only divided into the training set because it doesn't have reversed version as hard negative sample. C1 and C2 have different semantics and the reversed video is also reasonable and meaningful. Both of them are thus retained and annotated.

Next, for retained videos with meaningful strong temporal nature, annotators proceed to write fine-grained descriptions for them. Since we applied activity as the query to search engine for video crawling, there may be multiple matching videos for a brief query focusing solely on temporal features, which consequently leads to the occurrence of false negatives and diminishing the effectiveness of the evaluation [44, 55]. In order to mitigate this issue, annotators are required to describe not only the temporal features of videos but also their distinct spatial features. Each video is thus associated with a fine-grained annotation sentence, e.g. $(v_i, t_i)$ or $(\widetilde{v_i}, \widetilde{t_i})$. Finally, we obtain 21,537 videos paired with detailed descriptions. We show some examples in Figure 5.

**Rewriting for Diversity.** Previous work [9] has demonstrated that rewriting text descriptions for image-text contrastive learning can enhance the performance of CLIP [43]. Inspired by this, for

[4]using Appen Platform: https://www.appen.com/

the purpose of augmenting the diversity of text descriptions and facilitating effective video-text training, we provide GPT-4 with the human-written caption, and instruct it to rewrite nine extra sentences, requiring the rewritten sentences to exhibit diversity in sentence structure and vocabulary while retaining the key semantic information of the original sentence. Specifically, for a video-text pair $(v_i, t_i)$ or $(\widetilde{v_i}, \widetilde{t_i})$, we get $\{(v_i, t_{ij})\}_{j=1}^{10}$ or $\{(\widetilde{v_i}, \widetilde{t_{ij}})\}_{j=1}^{10}$ after rewriting. Ultimately, RTime contains ~210k video-text pairs, an order of magnitude increase. Due to the uncertainty in the quality of GPT-generated captions, we only add these generated captions in the training set, and merely utilize manually generated captions with higher quality in validation and test set.

## 3.2 Dataset Statistics

Table 1 compares our RTime dataset with other video-text datasets. RTime contains a total of 21,537 videos, each with one manually annotated caption and nine GPT-4 generated captions. Among all the videos, 16,530 videos have their temporally harder-negative counterparts, accounting for 76.8%. RTime is comparable in dataset scale to mainstream evaluation datasets. Compared to SSV2-Label and SSV2-Template, videos in RTime cover a broader range, addressing the domain limitation in these datasets. The activity list for RTime construction covers a wide range of natural activities with strong temporality. Some activities (verb-noun combinations) and a word-cloud based on the distribution of verb phrases are illustrated in the supplementary material which shows more balanced distribution of verb phrases in RTime. More importantly, text sentence lengths in RTime are longer than other similar datasets, indicating that our text annotations are finer-grained.

## 3.3 Benchmark Tasks Definition

Video-text retrieval requires the model to search for videos based on text queries (Text-to-Video retrieval, T2V) or to retrieve semantically matching textual descriptions based on video queries (Video-to-Text retrieval, V2T). We split our RTime dataset into training, validation, and testing subsets, containing 18,537, 1,000, and 2,000

Table 1: Statistics of RTime and other datasets

| Dataset | Domain | #Video clips | #Sentences | Avg len(sec) | Avg sent len | Duration(h) |
|---------|--------|--------------|------------|--------------|--------------|-------------|
| MSR-VTT[58] | open | 10K | 200K | 15.0 | 9.3 | 40 |
| YouCook II[62] | cooking | 14K | 14K | 19.6 | 8.8 | 176 |
| DiDeMo[1] | Flickr | 27K | 41K | 6.9 | 8.0 | 87 |
| ActivityNet-Cap[24] | action | 100K | 100K | 36.0 | 13.5 | 849 |
| LSMDC[45] | movie | 118K | 118K | 4.8 | 7.0 | 158 |
| SSV2-Label[25] | ego-centric action | 171K | 111K | 4.0 | 6.6 | 190 |
| SSV2-Template[25] | ego-centric action | 171K | 174 | 4.0 | 6.0 | 190 |
| RTime (Ours) | open | 21K | 210K | 20.4 | 20.2 | 122 |

videos, respectively. Note that we ensure that the raw video and its reversed counterpart are in the same subset. We propose three evaluation settings to assess video-text retrieval models.

**Standard Video-Text Retrieval (RTime-Origin).** This setting is similar to other standard video-text retrieval benchmarks without harder-negatives. We only use raw videos $v_j$ with its human-written captions $t_j$ and exclude reversed videos $\widetilde{v_i}$ in the test set, thus the test set contains 1000 video-text $(v_j, t_j)$ pairs. We use the commonly adopted Recall at K (R@K) as our evaluation metrics, which reports the percentage of correctly retrieved samples in the top K retrieval results, and K = 1, 5, 10 is applied in our experiments. These metrics are used for both text-to-video and video-to-text retrieval.

**Video-Text Retrieval with Harder-Negative Samples (RTime-Hard).** In this setting, we use both raw video and its reversed counterparts with their human-written captions. The inclusion of reversed videos places higher demands on models to possess comprehensive understanding of both temporal and spatial information. We use it as the primary setting to assess the performance of video-text retrieval models. We apply R@1, R@5, R@10 evaluation metrics for both text-to-video and video-to-text retrieval.

**Binary Video-Text Retrieval (RTime-Binary)** This task setting specifically evaluates the temporal understanding capability of models, including both the binary text-to-video retrieval and binary video-to-text retrieval. For binary text-to-video retrieval, given a text query, the model needs to select the correct video from the two candidate videos that have the same visual appearance but opposite temporal semantics. Similarly, for binary video-to-text retrieval, given a query video, the model needs to find the correct description from the two candidate descriptions. We use accuracy (Acc) as our evaluation metric in this setting, and a random selection yields an accuracy of 50%.

## 4 EMPIRICAL STUDY ON RTIME

In this section, we carry out extensive empirical studies on the proposed benchmark tasks with RTime to gain a more in-depth understanding of challenges in video-text retrieval. We first introduce the model architecture and learning strategy (Sec. 4.1). Next we evaluate and analyze a variety of video-text retrieval methods on RTime benchmark (Sec. 4.2). Furthermore we investigate the factors that could impact the temporal understanding capability of models in RTime-Hard and RTime-Binary tasks (Sec. 4.3), and finally we present some additional ablation analysis (Sec. 4.4) and qualitative results (Sec. 4.5).

### 4.1 Model Architecture and Learning Strategy

We use model with the two-stream architecture, which consists of a separate visual encoder and text encoder, followed by a cross-modal alignment module. The video and text encoders encode videos and texts into visual and textual features, respectively. The cross-modal alignment module involves a light transformer layer to fuse visual and textual features and output the similarity score matrix between video and text.

The learning objectives include the visual-textual contrastive loss (VTC) [3, 43] and the visual-textual matching loss (VTM) [25, 28, 30]. Specifically, given a batch of videos and texts' representations $(v_i, t_i)_{i=1}^{B}$ with size $B$, the VTC loss is computed as follows:

$$L_{\text{vtc}-v2t} = -\frac{1}{B}\sum_{i}^{B} \log \frac{\exp(t_i^\top v_i/\sigma)}{\sum_{j=1}^{B}\exp(t_i^\top v_j/\sigma)}, \tag{1}$$

$$L_{\text{vtc}-t2v} = -\frac{1}{B}\sum_{i}^{B} \log \frac{\exp(v_i^\top t_i/\sigma)}{\sum_{j=1}^{B}\exp(v_i^\top t_j/\sigma)}, \tag{2}$$

$$L_{\text{vtc}} = \frac{1}{2}(L_{\text{vtc}-v2t} + L_{\text{vtc}-t2v}), \tag{3}$$

where $\sigma$ is the temperature parameter. And the VTM loss is computed by:

$$L_{\text{vtm}-v2t} = \frac{1}{B}\sum_{i}^{B} CrossEntropy(y^{\text{vtm}}, p^{\text{vtm}}(v_i, t_{j\in\{i,i_{neg}\}})), \tag{4}$$

$$L_{\text{vtm}-t2v} = \frac{1}{B}\sum_{i}^{B} CrossEntropy(y^{\text{vtm}}, p^{\text{vtm}}(t_i, v_{j\in\{i,i_{neg}\}})), \tag{5}$$

$$L_{\text{vtm}} = \frac{1}{2}(L_{\text{vtm}-v2t} + L_{\text{vtm}-t2v}), \tag{6}$$

where $p^{\text{vtm}}$ denotes probability, $i_{neg}$ denotes a negative sample in the same batch, and $y^{\text{vtm}}$ denotes ground-truth label of matched or not. We use negative mining [30] for efficiency, sampling the negative samples in the same batch for VTM loss.

We also enhance the use of harder-negative samples in RTime by placing positive and harder-negatives (i.e. the reversed counterparts) in the same batch during fine-tuning, e.g. UMT-Neg denotes our fine-tuned UMT with such enhanced use of harder-negatives.

## 4.2 Benchmarking SOTA Models on RTime

We evaluate different SOTA models on the three RTime benchmark tasks: RTime-Origin, RTime-Both, and RTime-Binary, which require increasingly stronger temporal understanding capabilities. Specifically, we evaluate two image-text pre-trained models (i.e. CLIP [43] and BLIP [28]), one model pre-trained on video-text datasets with single frame (i.e. Singularity [25]), and two video-text models (i.e. VINDLU [8] and UMT [31]) in our zero-shot experiments. We also fine-tune two temporally-adapted image-text models based on CLIP (i.e. CLIP4Clip [38] and Ts2Net [35]) and UMT [31].

As shown in Table 2, since our video descriptions are more fine-grained and longer than previous benchmarks, they are less ambiguous and most models can achieve satisfactory results on RTime-Origin. But there is a performance gap between image-text pre-trained model (CLIP R@1 58.7, BLIP R@1 71.8), model pre-trained with one frame (Singularity R@1 74.1) and video-text pre-trained model (UMT R@1 80.3) with the same model size on RTime-Origin. The same trend is observed on RTime-Hard as well, with (CLIP R@1 28.8, BLIP R@1 36.2) vs (Singularity R@1 36.2) vs (UMT R@1 40.2). Significantly different from the results shown in Figure 2, where CLIP and Singularity perform comparably to UMT, these comparison experiments demonstrate that our RTime is more temporal-emphasized and therefore more effective in evaluating different video retrieval models. The above phenomenon is the same in larger-scale models (BLIP-L vs UMT-L).

Moreover, we observe that performances of various models on RTime-Hard all drop significantly compared to those on RTime-Origin, with the most significant performance drop in the R@1 metric. This is because models can easily find videos with the same visual appearance but different temporal semantics based on static visual information. However, further correctly distinguishing temporal order is challenging, resulting in R@1 degradation. We also observe that the performances of CLIP-based models (CLIP4Clip and TS2Net) are significantly improved after fine-tuning, which indicates to some extent the shortcomings of image-text models in temporal understanding, and also shows the necessity of learning temporal understanding ability for models.

The RTime-Binary task focuses only on evaluating the temporal understanding ability. Experimental results show that all models we adopt perform poorly, even after fine-tuning. This demonstrates that RTime indeed poses a significant challenge to current models. Even fine-tuned models can only achieve slightly better results than random selection. In addition, it is worth noting that UMT-Neg, which adopts our enhanced use of harder-negatives, achieves obvious gain on the RTime-Binary task.

Overall, the results of our benchmarking with various SOTA models on RTime tasks show that on the one hand, RTime indeed poses a higher challenge to video-text retrieval, on the other hand, current video-text models still fall short in temporal understanding.

## 4.3 Ablation on Temporal Understanding

We investigate factors that could possibly impact the temporal comprehension performance based on our UMT-Neg model. We conduct experiments under two task settings, RTime-Hard, which comprehensively assesses the spatio-temporal understanding capability of models, and RTime-Binary, which focuses on evaluating the temporal understanding ability.

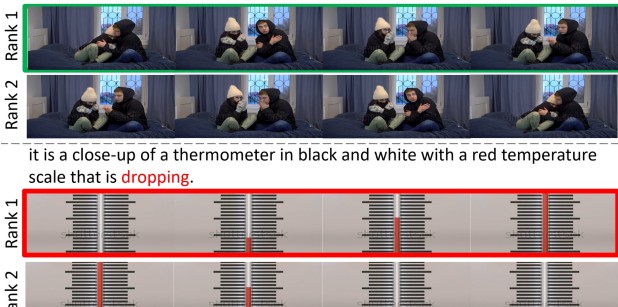

A man in a black jumper and a woman in a white hat embrace **before** the two rub their hands together to keep warm.

Rank 1

Rank 2

it is a close-up of a thermometer in black and white with a red temperature scale that is **dropping**.

Rank 1

Rank 2

**Figure 6: Success and failure cases of UMT in choosing the correct video given the text query. Green boxed represents the true answer while red boxed denotes the wrong answer.**

**Impact of leveraging harder negatives within same batch.** One of the prominent features of our RTime is that many video-text pairs have harder-negative samples (the reversed counterpart), which may be beneficial for the model to enhance its temporal understanding ability during fine-tuning. To ablate the impact of our enhanced use of harder-negatives (i.e. placing positive and harder-negatives in the same batch), we compare the performances of the fine-tuned model with (UMT-Neg) and without (UMT) such strategy. As shown in the last two lines in Table 2, UMT-Neg does demonstrate better temporal understanding, achieving relative improvements of +2.9% (R@1 46.3 vs 45.0) on RTime-Hard and +6.4% (Acc 54.5 vs 51.2) on RTime-Binary.

**Impact of number of input frames.** Learning of temporal information is theoretically closely related to the number of input frames used during fine-tuning. Our experimental results, shown in Table 3, demonstrate that increasing the number of input frames gradually improves the spatial-temporal understanding performance.

**Impact of temporal positional embedding.** Temporal positional embedding which contains frame position information plays a crucial role in the Transformer architecture for learning temporal information. As shown in Table 4, compared to results with spatial-only positional embeddings, it is evident that spatio-temporal positional embeddings are indeed beneficial for temporal understanding.

## 4.4 Additional Ablation Analysis

We further ablate other factors in data construction that may affect the video-text retrieval performance.

**Impact of rewriting in data construction.** From Table 5, comparing row 1 with row 3 as well as row 2 with row 4, we observe that applying "rewrite" strategy, which adds extra rewritten captions for each video into the training data, significantly enhances the model's spatio-temporal understanding, demonstrating its effectiveness.

**Impact of reverse in data construction.** From Table 5, comparing the row 1 with row 2 and row 3 with row 4, we observe that adding "reversed" videos and their text descriptions during training brings improvement, especially on temporal understanding. It is notable that with "rewrite" strategy, where more negative captions

**Table 2: Comparison of existing methods on the three RTime benchmarks. ZS: zero-shot, FT: Fine-Tuning. "-L" refers to models using ViT-L/14 as vision encoder and other the models using the ViT-B/16 with the same size.**

| | Method | RTime-Origin | | | | | | RTime-Hard | | | | | | RTime-Binary | |
| | | T2V | | | V2T | | | T2V | | | V2T | | | T2V | V2T |
| | | R@1 | R@5 | R@10 | R@1 | R@5 | R@10 | R@1 | R@5 | R@10 | R@1 | R@5 | R@10 | Acc | Acc |
|---|---|---|---|---|---|---|---|---|---|---|---|---|---|---|---|
| ZS | CLIP | 58.7 | 85.5 | 91.9 | 52.1 | 79.7 | 88.7 | 28.8 | 73.2 | 83.4 | 27.7 | 66.6 | 77.9 | 49.1 | 49.5 |
| | BLIP | 71.8 | 91.6 | 95.4 | 71.2 | 91.8 | 95.0 | 36.2 | 84.6 | 91.3 | 36.65 | 82.25 | 90.15 | 49.7 | 49.0 |
| | Singularity | 74.1 | 92.7 | 95.4 | 75.2 | 93.5 | 96.8 | 36.2 | 86.1 | 92.0 | 39.0 | 87.3 | 93.4 | 48.7 | 49.9 |
| | VINDLU | 80.2 | **96.4** | **98.8** | 79.9 | **97.2** | **98.7** | **41.1** | **91.9** | **95.9** | 41.3 | **92.3** | **96.9** | **50.9** | 49.9 |
| | UMT | **80.3** | 95.3 | 96.9 | **81.0** | 95.6 | 97.8 | 40.2 | 89.7 | 94.7 | **42.0** | 90.9 | 95.6 | 49.8 | **50.4** |
| | BLIP-L | 81.6 | 96.0 | 97.8 | 80.3 | 95.8 | 97.4 | 40.2 | 90.9 | 95.8 | 41.2 | 90.7 | 95.2 | 49.3 | 50.2 |
| | UMT-L | **84.7** | **97.8** | **98.9** | **85.7** | **97.9** | **99.3** | **45.4** | **94.1** | **97.4** | **44.8** | **94.6** | **98.3** | **53.1** | **51.0** |
| FT | CLIP4Clip | 75.2 | 94.7 | 97.7 | 75.5 | 95.0 | 97.8 | 37.3 | 88.0 | 94.0 | 37.3 | 88.6 | 93.7 | 49.8 | 49.8 |
| | Ts2Net | 74.3 | 95.3 | 97.4 | 76.3 | 95.6 | 97.8 | 37.8 | 88.8 | 94.8 | 38.9 | 88.5 | 94.9 | 50.7 | 50.0 |
| | UMT | **86.3** | 98.3 | **99.2** | 86.2 | 98.3 | 99.2 | 45.0 | 95.0 | **98.2** | 45.5 | 94.9 | 98.3 | 51.2 | 51.3 |
| | UMT-Neg (Ours) | 84.6 | 98.2 | 99.0 | 85.4 | **98.3** | **99.5** | **46.3** | **95.0** | 98.1 | **46.8** | **95.1** | **98.3** | **54.5** | **54.2** |

**Table 3: Performance comparison with different number of frames in fine-tuning. We input 12 frames during inference.**

| | RTime-Hard | | | | | | RTime-Binary | |
| | T2V | | | V2T | | | T2V | V2T |
| # | R@1 | R@5 | R@10 | R@1 | R@5 | R@10 | Acc | Acc |
|---|---|---|---|---|---|---|---|---|
| 1 | 38.1 | 86.0 | 92.5 | 36.1 | 83.6 | 91.3 | 49.9 | 50.6 |
| 4 | 40.9 | 90.0 | 95.8 | 38.5 | 89.1 | 95.0 | 51.4 | 49.7 |
| 8 | 42.8 | 91.3 | 96.1 | 41.9 | 91.2 | 95.4 | 51.7 | 51.9 |
| 12 | **46.3** | **95.0** | **98.1** | **46.8** | **95.1** | **98.3** | **54.5** | **54.2** |

**Table 4: Performance comparison with or without temporal positional embedding. PE: positional embedding.**

| | RTime-Hard | | | | | | RTime-Binary | |
| | T2V | | | V2T | | | T2V | V2T |
| PE | R@1 | R@5 | R@10 | R@1 | R@5 | R@10 | Acc | Acc |
|---|---|---|---|---|---|---|---|---|
| ✗ | 38.1 | 86.0 | 92.5 | 36.1 | 83.6 | 91.3 | 49.9 | 50.6 |
| ✓ | **46.3** | **95.0** | **98.1** | **46.8** | **95.1** | **98.3** | **54.5** | **54.2** |

**Table 5: Impact of certain processing strategies in data construction. RW: Rewrite, RV: Reverse**

| | | RTime-Hard | | | | | | Binary | |
| | | T2V | | | V2T | | | T2V | V2T |
| RW | RV | R@1 | R@5 | R@10 | R@1 | R@5 | R@10 | Acc | Acc |
|---|---|---|---|---|---|---|---|---|---|
| ✗ | ✗ | 43.35 | 93.8 | 97.5 | 44.4 | 93.7 | 97.4 | 51.6 | 51.8 |
| ✗ | ✓ | 44.5 | 93.9 | 98.1 | 44.7 | 93.7 | 97.6 | 52.9 | 51.4 |
| ✓ | ✗ | 45.3 | 94.6 | 97.8 | 44.2 | 94.3 | 97.8 | 52.2 | 51.1 |
| ✓ | ✓ | **46.3** | **95.0** | **98.1** | **46.8** | **95.1** | **98.3** | **54.5** | **54.2** |

**Table 6: Impact of test-set scale on performance.**

| | RTime-Hard | | | | | | RTime-Binary | |
| | T2V | | | V2T | | | T2V | V2T |
| scale | R@1 | R@5 | R@10 | R@1 | R@5 | R@10 | Acc | Acc |
|---|---|---|---|---|---|---|---|---|
| 1K | 47.3 | 95.2 | 97.7 | 47.6 | 96.8 | 98.5 | 53.7 | 53.7 |
| 2K | 46.3 | 95.0 | 98.1 | 46.8 | 95.1 | 98.3 | 54.5 | 54.2 |

set size, we also extract a subset of size 1,000 from RTime-Hard test set. As shown in Table 6, performance across different scales does not vary much, which confirms that the challenge of RTime-Hard does mainly come from more challenging temporal understanding.

## 4.5 Qualitative Results

We visualize some success and failure cases in choosing the correct video given the text query in Figure 6. In the top success case, UMT can correctly recognize that "embrace" occurs before "rub their hands," but in the bottom failure case, it fails to distinguish the "rise" and "fall" of the mercury column in the thermometer. Distinguishing temporal semantics in videos with identical visual appearance poses a higher challenge for existing models.

## 5 CONCLUSION

This work aims to address the lack of temporal understanding evaluation in existing video-text retrieval benchmarks. We introduce RTime, a novel fine-grained temporal-emphasized video-text dataset, carefully constructed in a top-down three-step pipeline by leveraging the power of large language models and human expertise. We further establish three benchmark tasks: RTime-Origin retrieval, RTime-Hard retrieval, and RTime-Binary retrieval, which can support comprehensive and faithful evaluation of video understanding capabilities especially in terms of temporal understanding. The extensive experiment analysis confirms that our RTime indeed poses higher challenges to video-text retrieval. We hope that our work will draw more attention to the importance of temporal understanding and contribute to more broad advancement of video-language understanding tasks such as video captioning, videoQA and Multimodal Large Language Model evaluation.

exists, the gain in RTime-Binary is more significant. This observation suggests the necessity for more temporal harder-negatives to enhance models' temporal understanding ability.

**Impact of test set scales on performance.** Since the RTime-Hard test set (2000) is twice the size of RTime-Origin (1000), one might argue that, in addition to the difficulty of temporal understanding itself, the significantly lower performance on RTime-Hard might also relate to the test set size. To verify the influence of test

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
