# OpenReview forum: "Reversed in Time: A Novel Temporal-Emphasized Benchmark for Cross-Modal Video-Text Retrieval"
_acmmm.org/ACMMM/2024/Conference — MM2024 Poster_

### Official Review · Reviewer_PhbB · 2024-05-23

**Rating:** 6
**Confidence:** 4

**Summary:**

The paper introduces RTime, a novel dataset for video-text retrieval that addresses shortcomings in existing benchmarks and emphasizes temporal understanding. The authors propose three retrieval benchmark tasks based on RTime and demonstrate its effectiveness in posing new challenges to video-text retrieval. Overall, the work presents a significant contribution to the field and provides valuable insights for advancing multimodal understanding research.

**Strengths:**

1. The proposal to construct the RTime dataset and establish new benchmarks for video-text retrieval, with an emphasis on temporal understanding, is a significant contribution to addressing deficiencies in current datasets.

2. The paper effectively presents the methodology used for constructing the RTime dataset in a clear and systematic manner, enabling readers to understand the process of dataset creation and the rationale behind the design choices. This clarity enhances the reproducibility and potential adoption of the proposed benchmark in the research community.

**Limitations:**

1.If some experimental results of the visual foundation model could be added, the results would be more convincing, and also provide more insights for research based on the foundation model.

2.The references [29] and [30] in the paper appear to be duplicated and should be revised accordingly.

**Suitability:**

3

---

### Official Review · Reviewer_KrzE · 2024-05-24

**Rating:** 3
**Confidence:** 4

**Summary:**

This paper proposed a new dataset which addresses the problem of lacking temporal dependency on existing datasets for video-text retrieval tasks by adding meaningful reversed videos. On the new dataset the zero-shot image-text pretrain models without temporal dependency are faced with performance degradation.

**Strengths:**

The new dataset is more challenging as large-scale image-text pretrain models cannot achieve competitive results. Reaching good performance requires efforts on temporal dependency learning.

**Limitations:**

1. The reversed videos as samples in the proposed dataset are wired. Some actions in the reversed videos obviously violate the physical rules. I wonder if the models trained on this dataset can learn from such abnormalities rather than the action itself to make predictions. I think a better way to emphasize the temporal dependency is to increase the length of the videos.
2. The references need improvements.

**Suitability:**

2

---

### Official Review · Reviewer_nGnc · 2024-05-24

**Rating:** 5
**Confidence:** 3

**Summary:**

This paper proposes RTime, a new temporal-emphasized video-text retrieval benchmark dataset, along with three new tasks: RTime-Origin, RTime-Hard, and RTime-Binary retrieval. This paper constructs the RTime dataset with a focus on temporal understanding by including pairs of videos with opposite temporal semantics as "harder negatives". They propose three new video-text retrieval tasks based on RTime to comprehensively evaluate models' temporal understanding capabilities. The extensive experiments show that RTime poses higher challenges to current models compared to existing datasets, especially for temporal understanding.

**Strengths:**

(1) Novelty: This paper creates a new dataset focused on temporal understanding for video-text retrieval is novel and valuable, as existing datasets lack emphasis on this important aspect. Including temporally reversed "harder negative" video pairs is a creative way to increase the difficulty for temporal reasoning.

(2) Technical approach: The top-down 3-step construction pipeline leveraging large language models (e.g. GPT-4) and human expertise is well-designed and effective for building the desired temporal-focused dataset. The use of LLMs improves efficiency while human verification ensures data quality.

(3) Evaluation: The paper conducts comprehensive experiments evaluating various state-of-the-art models on the proposed three tasks, providing in-depth analysis on different factors affecting temporal understanding performance. The results validate the higher challenge RTime poses for this aspect.

**Limitations:**

(1) Lack of comparison to video-retrieval datasets: The paper does not compare the proposed methods on other video-retrieval datasets. We cannot directly compare the results of the introduced dataset and other datasets. A discussion distinguishing different datasets would be beneficial for understanding the advantages of the introduced dataset.

(2) Lack of qualitative error analysis: The paper reproduces some different video retrieval methods on the introduced datasets.These methods provide different performance. Further analysis of the performance difference may be helpful.

**Suitability:**

2

---

### Official Review · Reviewer_BmxU · 2024-06-02

**Rating:** 4
**Confidence:** 4

**Summary:**

The paper proposes a new video-text retrieval dataset, RTime, which emphasizes the importance of temporal understanding. By creating positive and negative sample pairs of videos with significant temporal significance, the model's discriminative ability in the temporal dimension is enhanced. A top-down three-step construction scheme was adopted, combined with large-scale language models (such as GPT-4) and manual review, to generate a high-quality dataset containing 21k videos and 210k video text pairs, providing abundant resources for video text retrieval tasks. Based on the RTime dataset, three retrieval benchmark tasks were defined: RTime Origin, RTime Hard, and RTime Binary, which gradually increased the evaluation requirements for the model's time understanding ability. Through extensive experimental analysis, it has been demonstrated that the RTime dataset poses new challenges to existing models in evaluating video text retrieval models, especially in terms of temporal understanding, and promotes further research on video language understanding tasks. The paper also conducted empirical research on the factors that affect the model's ability to understand time, and presented some qualitative results. These analyses help to understand the limitations of the current model in terms of time and provide guidance for future research directions.

**Strengths:**

1. The structure of this paper is clear and easy to follow.
2. Through extensive experiments and detailed analysis, the shortcomings of existing models in temporal understanding were demonstrated, and the effectiveness of the RTime dataset in improving evaluation difficulty and promoting model development was verified.
3. The author promises to release the RTime benchmark, which will provide a valuable resource for the research community to promote progress in video text retrieval and multimodal understanding research.
4. Experimental results indicate that even state-of-the-art models face challenges on the RTime dataset, emphasizing the necessity of further research in the field of video text retrieval.

**Limitations:**

1. Although the RTime dataset has a moderate scale, its diversity may be limited compared to some existing large-scale datasets, and there are no corresponding statistical indicators to prove it in the paper.
2. Tab1 lacks comparison with several common datasets, such as Vatex and MSVD;
3. In L375-377, it is mentioned that the videos in the dataset are obtained from internet, but there is a sensitive issue: how to solve the copyright issue of these videos obtained from the internet? And how to ensure the long-term validity of these videos? That is to say, after the author release the dataset, can they ensure that the videos in the dataset can be effectively obtained by subsequent researchers in the long term?

**Suitability:**

3

---

### Meta-Review · Area_Chair_Zr1Y · 2024-06-30

**Recommendation:** Accept (Poster)
**Confidence:** 4

**Metareview:**

This paper presents a new video-text retrieval dataset called RTime. This dataset is designed focusing on temporal understanding.
The authors conducted experiments to evaluate state-of-the-art models on the newly proposed three tasks and show the new chalenges.
The dataset is carefully designed and it will be releast as a list of video URL.
This paper would give a great impact to our MM community and therefore I would recommend acceptance of this paper.